# Impacts of Climate Change and Land Use/Cover Change on the Net Primary Productivity of Vegetation in the Qinghai Lake Basin

**DOI:** 10.3390/ijerph20032179

**Published:** 2023-01-25

**Authors:** Jinlong Zhang, Yuan Qi, Rui Yang, Xiaofang Ma, Juan Zhang, Wanqiang Qi, Qianhong Guo, Hongwei Wang

**Affiliations:** 1Gansu Provincial Key Laboratory of Remote Sensing, Hei’he Remote Sensing Experimental Research Station, Northwest Institute of Eco-Environment and Resources, Chinese Academy of Sciences, Lanzhou 730000, China; 2Xining Comprehensive Survey Center of Natural Resources, China Geological Survey, Xining 810000, China; 3Bureau of Natural Resources of Zhuoni, Zhuoni 747600, China

**Keywords:** net primary productivity, land use/land over, spatio-temporal variability, driving factors, Qinghai Lake Basin

## Abstract

The Qinghai Lake Basin acts as a natural barrier, preventing the western desert from spreading eastward. This is an important link in preserving the ecological stability of the northeastern region of the Qinghai–Tibet Plateau (QTP). Therefore, quantitative research into the net primary productivity (NPP) of vegetation and its driving force in the Qinghai Lake Basin is required. The effects of land use/cover change (LUCC) and climate change on NPP in the Qinghai Lake Basin were studied using R-contribution ratio and partial correlation analysis methods using MOD17A3H products, Land Use/Land Cover (LULC) data, and meteorological data. (1) The LULC of the Qinghai Lake Basin showed a trend that “the area of grassland, cultivated land, and unused land continued to decrease, while the area of other LULC types increased” from 2000 to 2020, according to the study’s findings. Grassland, water bodies, construction land, and unused land dominated the mutual transformation of LULC types. (2) The NPP of the basin showed a growing trend, with a growth rate of 3.93 gC·m^–2^·a^–1^ before 2010 and 0.88 gC·m^–2^·a^–1^ after 2010. Significant regional heterogeneity was found in NPP, with gradients decreasing from southeast to northwest. (3) The impact of LUCC on overall NPP changes had gradually increased. Climate change has been the primary driver of NPP changes in the Qinghai Lake Basin over the last 20 years.

## 1. Introduction

Carbon dioxide (CO_2_) emissions caused by industrial development have broken the original carbon balance and acted on climate change, which has significantly impacted the structure, function, and productivity of terrestrial ecosystems [1,2,3,4]. In the face of a slew of environmental issues, including global warming, CO_2_ concentration rise, sea level rise, and land use/cover change (LUCC), the study of the carbon cycle of terrestrial ecosystems and their interactions with the environment serves as the theoretical foundation for promoting ecosystem balance, mitigating, and responding to global change [5,6]. The accumulation of dry plant materials in a given time and space is defined as net primary productivity (NPP) [7]. As an important component of the study of carbon cycling, NPP reflects the capacity of vegetation production under natural conditions and is also a key factor in terrestrial ecosystems’ responses to global change [8,9,10,11].

In the context of global change, spatial and temporal evolution and the driving mechanism of NPP are important topics in current NPP research. The study found that vegetation NPP is influenced by many factors, including temperature, precipitation, topography, soil, CO_2_, and human activities, and that the influencing factors vary greatly across regions [12,13,14]. Although scholars have conducted a lot of research on the driving force of NPP [15,16,17,18,19,20,21,22], they mostly choose climate factors, among which temperature and precipitation are the most commonly chosen. At the same time, there are relatively few studies on the influence of terrain, soil, LUCC, and human activities on NPP. With the impact of human beings on the ecosystem through the development and utilization of land resources, the global carbon balance has changed directly or indirectly. The influence of LUCC on NPP has become an important part of the current carbon cycle research [23]. They concentrated on the effect of one or both of these factors on NPP while ignoring the effect of multiple factors. Furthermore, while previous research on the temporal and spatial variation and influencing factors of NPP have covered different spatial scales, such as global, national, and regional [24,25,26,27], the time scale is relatively single and the characteristics of NPP-phased changes are ignored [28]. Therefore, studying the spatial-temporal variation of NPP and quantitatively analyzing the relationship between NPP and LUCC, climate change, and other factors, as well as revealing the main influencing factors of NPP change in different periods, to understand the driving mechanism of NPP evolution, ecological environment protection, and sustainable development planning formulation, has important theoretical and practical implications [29,30].

The Qinghai Lake Basin is critical as an ecological security barrier in preventing western desertification to the east and sustaining ecological security in the northeast of the Qinghai–Tibet Plateau (QTP). In recent years, research on NPP in the Qinghai Lake Basin has primarily focused on the spatial and temporal variation of NPP and its response to climate change [31,32,33]. Based on this, the following are the primary research objectives of this paper: (1) to investigate the spatial and temporal evolution characteristics of NPP and Land Use/Land Cover (LULC) in the Qinghai Lake Basin and (2) to quantitatively investigate the response relationship between NPP and LUCC, climate, and other factors over time. The results will not only help to master the ecosystem quality and natural production capacity of the Qinghai Lake Basin but also provide support for rational decision-making regarding LUCC and ecological environmental governance and planning and provide a scientific basis for the coordination of economic development and environmental protection in the area.

## 2. Materials and Methods

### 2.1. Overview of the Study Area

The Qinghai Lake Basin is located in the northeastern part of the QTP (36°15′–38°20′ N, 97°50′–101°20′ E) and is bounded to the north bordered by the Datong Mountain, to the south by the Qinghai Nanshan Mountain, and to the west by the Qaidam Basin (Figure 1). An inland basin enclosed by mountains may be found to the east of the Riyue Mountain. The administrative division spans four counties in three different states of the Qinghai Province, including the Republican County in the Hainan Tibetan Autonomous Prefecture, Gangcha County and Haiyan County in the Haibei Tibetan Autonomous Prefecture, and Tianjun County in the Haixi Mongolian Tibetan Autonomous Prefecture. The combined area of these counties is approximately 29,600 km^2^. The geography of the basin climbs from southeast to northwest, with an average elevation of approximately 3717 m dominated by mountains. The winter season is brief and long, with dry conditions, little rainfall, cold, windy weather, and an overall typical plateau continental climate. Because of the unique geographic setting and climatic conditions, the grassland has evolved into a dominant component of the regional ecosystem in the basin and is now a significant source of high-quality pasture and animal husbandry products. Since it is located at the confluence of China’s eastern monsoon humid zone and western desert region, the Qinghai Lake Basin is extremely vulnerable to local and regional environmental changes.

### 2.2. Data and Preprocessing

#### 2.2.1. Net Primary Productivity (NPP) Data

The NPP data for 2001–2020 were obtained from the Moderate Resolution Imaging Spectroradiometer (MODIS) data product MOD17A3H Version 6 from the U.S. Geological Survey (http://lpdaac.usgs.gov (accessed on 13 May 2022)), which has a spatial resolution of 500 m and a temporal resolution of 1 a. Splicing, conversion projection, and cutting raw data were used to obtain the NPP data from the Qinghai Lake Basin. Based on this information, the Sen-Mann–Kendall method and the partial correlation analysis method were used to investigate the spatial and temporal variation characteristics of NPP and its response to climatic factors.

#### 2.2.2. Land Use/Land Cover (LULC) Data

The 24 downloaded images were preprocessed using radiometric calibration and atmospheric correction using ENVI software based on the Landsat thematic mapper (TM) image. The LULC data from the Qinghai Lake Basin in 2000, 2010, and 2020 were interpreted in ArcGIS 10.3 using human–computer interaction. The data have a spatial resolution of 30 m and a correctness rate of interpretation of more than 85%. On the basis of this information, the spatial and temporal distribution characteristics of LULC, as well as its response to NPP in the Qinghai Lake Basin, were investigated using the LUCC dynamic degree and R-contribution ratio.

#### 2.2.3. Meteorological Data

The China National Qinghai-Tibet Plateau Science Data Center (http://data.tpdc.ac.cn (accessed on 20 May 2022)) provided monthly average temperature and monthly precipitation datasets (2000–2020) with a geographical resolution of approximately 1 km [34]. The meteorological data were sampled to a spatial resolution consistent with the NPP to maintain data consistency. A partial correlation analysis of these data with NPP was performed to investigate the effect of climate on NPP.

#### 2.2.4. Other Data

The National Qinghai-Tibet Plateau Scientific Data Center (http://data.tpdc.ac.cn (accessed on 25 May 2022)) provided information on permafrost distribution in the Qilian Mountain [35]. The data were used to examine the impact of various permafrost conditions on NPP.

### 2.3. Method

#### 2.3.1. Land Use and Cover Change (LUCC) Analysis

(1)Dynamic degree

The dynamic degree of LUCC is divided into two subcategories: dynamic single and dynamic comprehensive LULC. The former reflects the rate of change of a specific type of LUCC over time, while the latter reflects the overall rate of change of regional LUCC over time [36]. Their respective calculation formulas are provided below:(1)$$K=\frac{U_{b}-U_{a}}{U_{a}}\times \frac{1}{T}\times 100\%$$
(2)$$\mathrm{LC}=\left(\frac{\sum_{i=1}^{n}\Delta LU_{i-j}}{2\sum_{i=1}^{n}LU_{i}}\right)\times \frac{1}{T}\times 100\%$$
where *K* is the single LUCC dynamic attitude; *LC* is the integrated LUCC dynamic attitude; *U_a_* and *U_b_*, respectively, *r* to the area of a certain LULC type at the beginning and end of the study period; *LU_i_* refers to the area of the ith LULC type at the beginning of the study period; and Δ*LU_i–j_* refers to the absolute value of the sum of the area of the number. *i* refers LULC type transformed into other LULC types during the study period; *n* refers to the number of LULC types; and *T* refers to the length of the study period.

(2)Chord Diagram

The trajectory map of LUCC migration is a visual representation based on the LUCC transition matrix [37]. The graph includes arcs and chords, where the arcs represent various LULC types and the chords represent the transformation of various LULC types [38].

(3)R-contribution Ratio

The method proposed by Hicky et al., was used to calculate the impact amount and contribution rate of LUCC and climate change on the total NPP of different LULC types, respectively [39,40]. Considering cultivated land as an example, if *S*_1_, *S*_2_, *NPP*_1_, and *NPP*_2_ express the LULC area and NPP per unit area, respectively, in times *t*_1_ and *t*_2_, the variation of total NPP (∆*NPP*_T_) should be expressed as Equation (3):(3)$$\Delta NPP_{T}=NPP_{2}\times S_{2}-NPP_{1}\times S_{1}$$

Since $\Delta S=S_{2}-S_{1}$, $\Delta NPP=NPP_{2}-NPP_{1}$, Equation (3) becomes:(4)$$\Delta NPP_{T}=\left(NPP_{1}+\Delta NPP\right)\times \left(S_{1}+\Delta S\right)-NPP_{1}\times S_{1}=\Delta NPP\times S_{1}+\Delta S\times NPP_{1}+\;\Delta S\times \Delta NPP$$
where, $\Delta NPP\times S_{1}$ is the impact amount of climate change on the total NPP of cultivated land; $\Delta S\times NPP_{1}$ is the impact amount of LUCC on the total NPP of cultivated land; and ΔS×ΔNPP is their common interaction on the total NPP of cultivated land.

Not considering their common interactions, the following equations show the contribution rate in climate variables (Equation (5)) and LUCC (Equation (6)) on total NPP (*R*_1_, *R*_2_).
(5)$$R_{1}=\frac{\left|\Delta NPP\times S_{1}\right|}{\left|\Delta NPP\times S_{1}\right|+\left|\Delta S\times NPP_{1}\right|+\left|\Delta S\times \Delta NPP\right|}\times 100\%$$
(6)$$R_{2}=\frac{\left|\Delta S\times NPP_{1}\right|}{\left|\Delta NPP\times S_{1}\right|+\left|\Delta S\times NPP_{1}\right|+\left|\Delta S\times \Delta NPP\right|}\times 100\%$$
where, Δ*S* and Δ*NPP* are the area variation of LULC and NPP change at a period, respectively.

#### 2.3.2. Sen-Mann–Kendall Method

Sen et al., (1968) proposed the Sen trend degree analysis method in 1968, which is a technique for examining long-term trends by computing the median value of the sequence, effectively reducing noise interference [41]. This value was calculated using the following equation:(7)$$\rho =\mathrm{median}\frac{x_{j}-x_{i}}{j-i},\;1<i<j<n$$
where *x_i_*, *x_j_* are the NPP time series; ρ is the slope; *ρ* < 0 denotes a downward trend in the time series NPP, whereas *ρ* > 0 denotes an increasing trend in NPP. The Mann–Kendall method requires no knowledge of the sequence distribution and can effectively reduce the impact of missing time series data and outliers on the research results. Therefore, this method is introduced to complete the Sen series distribution significance test [42,43,44].

When *n* < 10, the statistic S was used directly for the bilateral trend test.
(8)$$S=\overset{n=1}{\underset{i=1}{\sum }}\overset{n}{\underset{j=i+1}{\sum }}\mathrm{sign}\left(x_{j}-x_{i}\right),\;\;\;\mathrm{sign}\;\left(x_{j}-x_{i}\right)=\left\{\begin{array}{c} 1\;\;\;\left(x_{j}-x_{i}\right)>0\\ 0\;\;\;\left(x_{j}-x_{i}\right)=0\\ -1\;\left(x_{j}-x_{i}\right)<0 \end{array}\right.$$

The statistic *S* approximately obeyed the standard normal distribution when *n* ≥ 10, whereas the statistic *S* approximates the conventional normal distribution. At the *α* confidence level, if |Z| ≥ 1.96, it passes the significance test with a 95% confidence level [45].
(9)$$\mathrm{Var}\left(S\right)=\frac{n\left(n-1\right)\left(2n+5\right)-\sum_{i=1}^{m}t_{i}\left(t_{i}-1\right)\left(2t_{i}+5\right)}{18}$$
(10)$$Z=\left\{\;\begin{array}{c} \frac{S-1}{\sqrt{var\left(S\right)}}\;\;\;S>0\\ \;\;\;\;\;\;0\;\;\;\;\;\;\;\;\;\;S=0\\ \frac{S+1}{\sqrt{var\left(S\right)}}\;\;\;S<0 \end{array}\right.$$
where *n* represents the amount of data in the sequence; m represents the number of knots (repeated data groups) in the sequence; and *t_i_* represents the width of knots (the number of repeated data in the number *i* group of repeated data groups).

#### 2.3.3. Partial Correlation Analysis

The relationship between vegetation NPP, temperature, and precipitation was investigated using the partial correlation approach in the Qinghai Lake Basin [29].
(11)$$R_{xy,z}=\frac{R_{xy}-R_{xz}R_{yz}}{\sqrt{\left(1-R_{xz}^{2}\right)}\sqrt{\left(1-R_{yz}^{2}\right)}}$$

In the formula, after controlling for the independent variable—temperature (precipitation)—*R_xy,z_* is the partial correlation coefficient between the dependent variable NPP and the independent variable precipitation (temperature). A value less than 0 denotes a negative correlation, while a value greater than 0 denotes a positive correlation. Furthermore, the greater the absolute value of R, the stronger the correlation.

## 3. Results

### 3.1. Spatio-Temporal Variation of Land Use/Land Cover (LULC) in Qinghai Lake Basin from 2000 to 2020

From 2000 to 2020, the majority of LULC types in the Qinghai Lake Basin were grasslands, water bodies, and unused land, accounting for 62.39–62.69%, 17.99–18.39%, and 15.61–15.98%, respectively (Figure 2). The presence of grasslands in the region was the most significant of the three LULC types. The main water body in the southeast is Qinghai Lake, with a small amount of unused land to the northwest of the basin and along the east bank of Qinghai Lake. The amount of woodland and cultivated land was small, accounting for 1.59–1.65% and 1.51–1.52%, respectively. The former was concentrated in the northern region of Gangcha County, while the latter was concentrated along the north bank of Qinghai Lake. The construction land area was determined to be the smallest, accounting for approximately from 0.07% to 0.45% of the total area and is spread out along the west and north banks of Qinghai Lake.

From 2000 to 2020, the area of grasslands, cultivated land, and unused land in the the Qinghai Lake Basin decreased, while the area of other types increased. From 2000 to 2010, the overall dynamic degree of LUCC increased, with 7.78% representing the fastest growth rate (Table 1). The grassland area has decreased by 85.58 km^2^ over the last 21 years, while the cultivated land area has decreased by only 4.29 km^2^. From 2010 to 2020, the area of woodland increased steadily, with a total increase of 17.85 km^2^ and a faster growth rate of 0.37%. Before 2010, the water area decreased by 45.57 km^2^ (0.08%) and then increased by 116.53 km^2^ (0.22%), indicating a fluctuating but increasing trend. From 2000 to 2010, the construction land area increased by 90.51 km^2^ at a rate of 42.32%. The area growth slowed after 2010, with a 1.81% growth rate. Between 2000 and 2020, the area of unused land decreased by 109.08 km^2^ and by 104.92 km^2^ between 2010 and 2020, at a rate of 0.22%.

### 3.2. Analysis of Land Use and Cover Change (LUCC) in Qinghai Lake Basin from 2000 to 2020

The mutual change in grassland, construction land, water bodies, and unused land in the Qinghai Lake Basin over the last 21 years was discovered to be representative of the LULC shift of the region (Figure 3). Between 2000 and 2010, a total of 2205.19 km^2^ of LULC was performed, accounting for approximately 7.43% of the overall area of the basin. With a transfer area of 69.62 km^2^ and 265.18 km^2^, respectively, there was more LULC transported out than in the case of both grassland and water bodies. The construction land transfer was greater than the transfer out. The transfer is primarily caused by grassland; the transfer in and out of other LULC types was considered fairly balanced. The total area of LULC transfer from 2010 to 2020 was determined to be 788.57 km^2^, accounting for 2.66% of the total area of the watershed. The transfer out of grassland was greater than the transfer in, with transfer areas of 114.89 km^2^ and 23.77 km^2^, respectively. It was primarily transformed into water bodies and construction land. The transfer out of the unused land was also greater than the transfer in, with the transfer areas comprising 151.39 km^2^ and 124.65 km^2^, respectively. It was mostly transformed into water bodies and grassland. The transfer of cultivated land remained fairly balanced. On the other hand, the transfer in of water bodies, woodland, and construction land was greater than the transfer out. The water bodies were mostly extracted from unused land and grassland, while the woodland and construction land came from cultivated land and grassland.

### 3.3. Analysis of Changes of Net Primary Productivity (NPP) of Qinghai Lake Basin

At various times, the NPP in the Qinghai Lake Basin displayed a pattern of low spatial distribution in the northwest and high spatial distribution in the southeast. From 2001 to 2010, the NPP in the area west of Tianjun County was less than 200 gC·m^–2^·a^–1^, accounting for 45.82% of the total basin area. The NPP was greater than 300 gC·m^–2^·a^–1^ in a few areas along the southern shore of Qinghai Lake and north of Gangcha County, accounting for 19.80% of the total area of the basin (Figure 4a-1). In 2010–2020, the proportion of the watershed area with NPP less than 200 gC·m^–2^·a^–1^ decreased to 37.80%, while the proportion of the watershed area with NPP greater than 300 gC·m^–2^·a^–1^ increased to 29.20%, with the distribution extending to the western and northern areas (Figure 4a-2). Between 2001 and 2010, the NPP of 97.52% of the basin areas increased, with 22.73% of those regions experiencing a significant increase (*p* < 0.05). This was primarily distributed in the northwestern and central areas of the basin. During this time, a portion of the regional NPP dispersed around Qinghai Lake decreased by 2.38% (Figure 4b-1). Only 1.63% of the regions in the northwest of the basin, the southern, and northeastern coastlines of the lake area showed a significant rise (*p* < 0.05) in the NPP from 2010 to 2020, whereas 65.83% of the regions showed an increase. During this time period, the proportion of the lake with a significant decline (*p* < 0.05) increased from 0.05% to 0.64%, with the majority of the affected areas concentrated in the northeastern and eastern regions of the lake (Figure 4b-2).

The inter-annual variation trend of the vegetation NPP in the basin showed an overall increase from 2001 to 2020, while the growth rate continued to decline. The NPP increased at a faster rate between 2001 and 2010, reaching 3.93 gC·m^–2^·a^–1^ (Figure 5). For this time period, the annual average NPP values remained between 186.58 and 247.53 gC·m^–2^·a^–1^, with a multi-year mean of 211.79 gC·m^–2^·a^–1^. From 2010 to 2020, the vegetation NPP exhibited a slow fluctuating growth trend with a growth rate of 0.88 gC·m^–2^·a^–1^. For this time period, the annual average value of NPP ranged from 210.59 to 267.27 gC·m^–2^·a^–1^, with a multi-year average value of 236.65 gC·m^–2^·a^–1^. The lowest NPP value was recorded in 2012 (210.59 gC·m^–2^·a^–1^), while the highest was recorded in 2014 (267.27 gC·m^–2^·a^–1^).

## 4. Discussion

### 4.1. Analysis of the Effect of Changes in Land Use/Cover Change (LUCC) on the Vegetation NPP

In terms of mean NPP values, cultivated land, woodland, grassland, and unused land were discovered to be the top four LULC types [28]. Given that the vegetation in the basin is primarily composed of shrubs with significant space between the plants, the NPP per unit area of woodland was slightly lower than that of cultivated land [46]. From 2001 to 2010, the average annual NPP of cultivated land, woodland, grassland, and unused land increased. After 2010, the average annual NPP of cultivated land continued to rise, while the NPP of other types began to fall (Table 2).

The impact amounts of climate change and LUCC on the total NPP were calculated according to Equations (3) and (4). The contribution rates of climate change and LUCC to the total NPP were calculated using Equations (5) and (6), respectively (Table 3). From 2001 to 2010, the LUCC contributed much less to the change in the various types of NPP than climate change and it controlled the increase in the various types of NPP (except woodland). In the period 2010–2020, the contribution of LUCC to the change in the various types of NPP had gradually increased, but the contribution of climate change to the NPP change was still greater than that of the LUCC. Grassland and unused land were both negatively impacted by climate change and LUCC, severely limiting the total NPP growth. Facing the problem of “the inhibitory effect of LUCC on the total growth of NPP”, the government departments should coordinate the proportion of LULC types in the regional resource allocation planning process to maintain the regional NPP. Finally, climate change was identified as the primary cause of the change in NPP in the Qinghai Lake Basin. Wang et al. (2022) reported temperature and human activities to be the two most important drivers of significant changes in grassland NPP in the Qinghai Province [47]. In contrast, Zhou et al. (2015) and Wang et al. (2016) discovered that human activities in Northwest China and the arid and semi-arid regions have a greater impact on NPP than climate change [48,49]. 

LUCC (except woodland) was found to have a deterrent effect on the rise of total NPP from 2001 to 2020. However, the NPP has continued to rise for two main reasons: First, because of the large base of grassland area, which accumulates the most organic matter, the unit area NPP increased by 51.76 gC·m^–2^·a^–1^, despite its area slightly decreasing by 85.58 km^2^ over the last two decades, contributing to an increase in the total NPP in the basin. Second, climate change had a significantly greater impact on the change in total NPP of each category in the basin than LUCC, which significantly offset the declining trend caused by LUCC, causing the total NPP to rise. This was consistent with the findings by Jia (2018) [50].

### 4.2. Analysis of the Effects of Climate Change on Vegetation Net Primary Productivity (NPP)

The climate of the Qinghai Lake Basin has shifted dramatically in recent decades [51]. The annual mean temperature and precipitation had positive correlations with NPP from 2001 to 2010, with correlation values of 0.578 and 0.447, respectively, when the effects of the temperature and precipitation on NPP were examined (Figure 6). However, none of them passed the threshold of *p* < 0.05. From 2010 to 2020, there was virtually no correlation between the NPP and annual mean temperature and precipitation.

The pixel scale investigated the partial relationship between the temperature, precipitation, and NPP. Approximately 15.14% of the total area showed a significant (*p* < 0.05) positive correlation between the NPP and air temperature from 2001 to 2010, which was found to be primarily concentrated in the northwest of the basin and the northern part of the Gangcha County (Figure 7a-1). Only 2.17% of the area showed a significant (*p* < 0.05) positive correlation from 2010 to 2020, with the majority of this region located on the north and northwest banks of Qinghai Lake (Figure 7a-2).

Nearly 1.99% of the total area showed a significant relationship between NPP and precipitation from 2001 to 2010 (Figure 7b-1). The regions with significant (*p* < 0.05) positive correlations accounted for 1.96% of the total area, which was mostly found in the western region of the basin and the northern portion of Gangcha County; the regions with significant (*p* < 0.05) negative correlations were scant, accounting for only 0.03%. However, the area where NPP was significantly (*p* < 0.05) positively associated with precipitation decreased to 1.86% in 2010–2020, which was concentrated in the northwest and northeast shore of the Lake, while the area with a significant (*p* < 0.05) negative correlation was nearly 0.07% (Figure 7b-2).

Finally, temperature has been identified as the primary climatic variable influencing NPP fluctuations in the Qinghai Lake Basin over the last 20 years. The temperature has a greater impact on vegetation NPP than precipitation, as reported by Guo et al. (2006), He et al. (2005), and Liang et al. (2015) [52,53,54]. Furthermore, this study found that air temperature and precipitation are primarily positively related to NPP and primarily stimulate the growth of NPP vegetation in the basin. This is similar to the research conclusion reached by Li et al. (2023) [55]. The primary factor promoting the significant increase in NPP during this period was determined to be temperature, as evidenced by the fact that the regions with a significant positive correlation between temperature and NPP were primarily distributed in regions where the NPP increased significantly from 2001 to 2010 [47]. Temperature, precipitation, and significant NPP connection areas were all found to be less than 3% from 2010 to 2020, indicating that climatic variables were not the primary causes of the significant variations in NPP during this time.

### 4.3. Analysis of Other Factors and Net Primary Productivity (NPP) of Qinghai Lake Basin

The NPP is affected not only by climate change and anthropogenic activities but also by permafrost, desertification, and other factors [56,57]. From 2001 to 2020, the interannual trends of NPP in permafrost and non-permafrost areas of the Qinghai Lake Basin were the same, indicating an overall increase (Figure 8). However, its rate of growth slowed after 2010. The mean values of NPP in non-perennial permafrost and perennial permafrost areas from 2001 to 2010 were found to be 288.70, 202.84 gC·m^–2^·a^–1^ and increased to 324.55 and 223.11 gC·m^–2^·a^–1^ from 2010 to 2020, respectively. In the last 20 years, both non-perennial and perennial permafrost areas have been found to be relatively well hydrated. However, because the former had much higher temperatures than the latter, the mean NPP values in the non-perennial permafrost area were higher than those in the perennial permafrost area. This is in line with the findings of Shen et al., (2015) [58].

The sandy area in the Qinghai Lake Basin decreased from 647.75 km^2^ to 470.70 km^2^ from 2001 to 2020, while the NPP increased from 128.51 gC·m^–2^·a^–1^ to 142.87 gC·m^–2^·a^–1^. It has been similarly reported that the area of desertification in the Qinghai Province has reversed since 2001 [59]. Climate change is an important factor responsible for the decrease in the sand area in the Qinghai Lake Basin. The climate of the region has been warming and humidifying over the past 20 years. The increase in temperature aids the speed of sand dune movement. However, the increase in precipitation increases the viscosity of sand particles, thereby effectively inhibiting the movement and change of sand dunes [60,61]. In addition, the effective implementation of projects such as “returning cultivated land to woodland and grassland” and “desertification land management” has not only improved the vegetation cover but also reduced regional desertification to a certain extent [62]. 

The impact factors discussed in this paper are generally restricted to temperature, precipitation, LUCC, permafrost, and desertification. More factors, such as sunshine hours, terrain factors, soil type, etc., will be added to analyze the impact of factors on NPP in the Qinghai Lake Basin. Furthermore, only annual scale analysis was used to assess the impact of climate change on NPP. There may be differences in the seasonal and monthly scales, which is a direction worth considering in the future.

## 5. Conclusions

The temporal and geographical evolution characteristics of NPP and LULC in the Qinghai Lake Basin were revealed using the LUCC analysis approach, Sen-Mann–Kendall method, and partial correlation method and the responses of the NPP to climate, LUCC, and other factors were investigated. The findings show that (1) grassland, water bodies, and unused land were the most common LULC types in the Qinghai Lake Basin. The overall change rate of LUCC tended to weaken, while the proportion of the transferred area of each LULC type decreased from 2000 to 2020. (2) The change in NPP demonstrated an overall increasing trend, with 22.73% of the regional NPP demonstrating a significant increase (*p* < 0.05) trend, mainly dispersed in the northwest and center of the basin before 2010, while the locations where the growth was significantly greater (*p* < 0.05) shrank to 1.63% after 2010. (3) It was determined that the impact of LUCC on the overall changes in total NPP in the basin was significantly less than that of climate change. The effect of temperature on NPP was greater than that of precipitation. The climate factors influenced the change in NPP in various frozen soil areas and sandy land.

## Figures and Tables

**Figure 1 ijerph-20-02179-f001:**
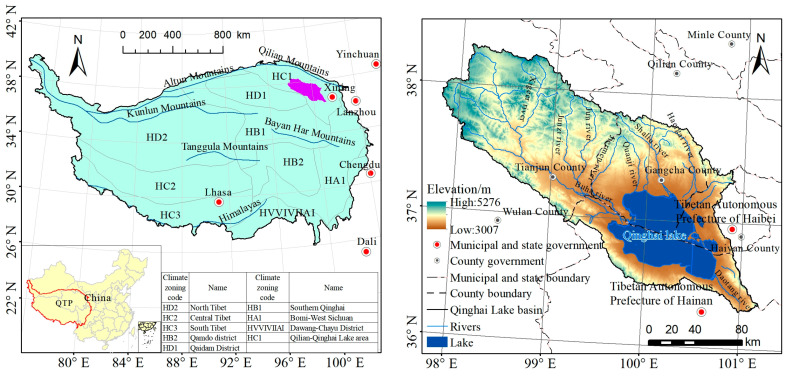
Geographical location map of the study area.

**Figure 2 ijerph-20-02179-f002:**
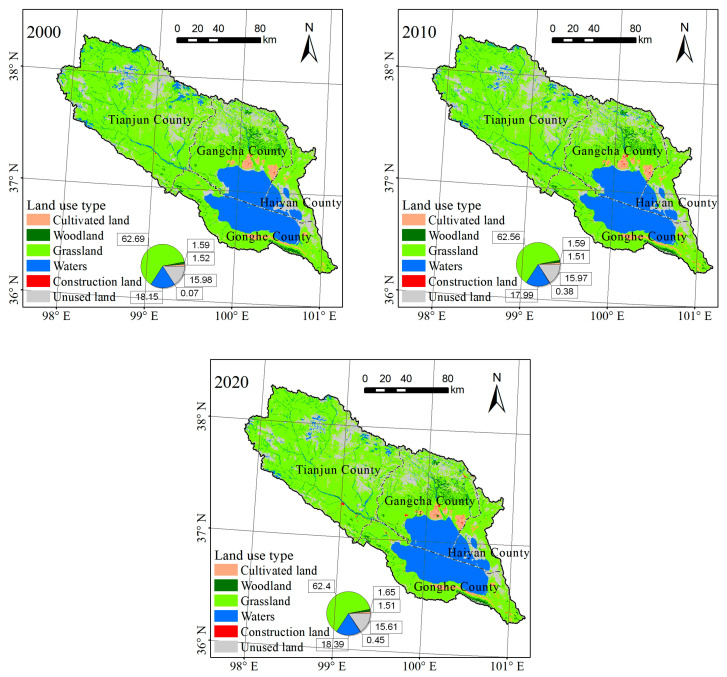
Spatial distribution of the land use/land cover (LULC) types in Qinghai Lake Basin from 2000 to 2020.

**Figure 3 ijerph-20-02179-f003:**
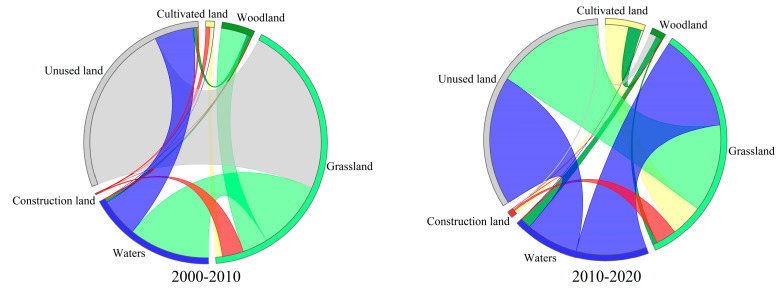
Land use and cover change (LUCC) in the Qinghai Lake Basin from 2000 to 2020 in a chord diagram.

**Figure 4 ijerph-20-02179-f004:**
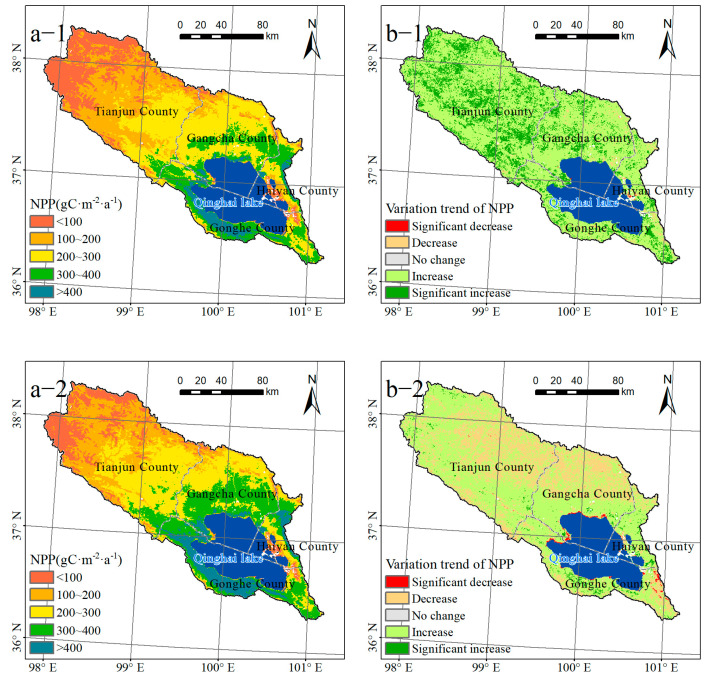
Spatial distribution (**a-1**,**a-2**) and change trend (**b-1**,**b-2**) of net primary productivity (NPP) in Qinghai Lake Basin from 2001 to 2020 (1 and 2 are 2001–2010 and 2010–2020, respectively).

**Figure 5 ijerph-20-02179-f005:**
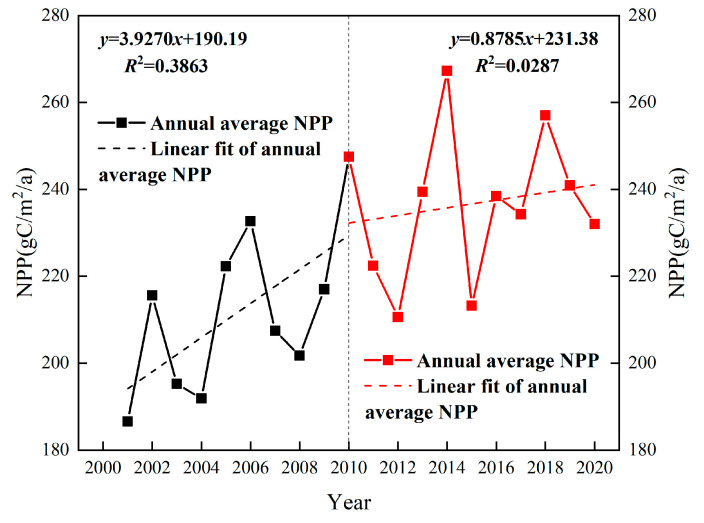
Trends in net primary productivity (NPP) in the Qinghai Lake Basin from 2001 to 2020.

**Figure 6 ijerph-20-02179-f006:**
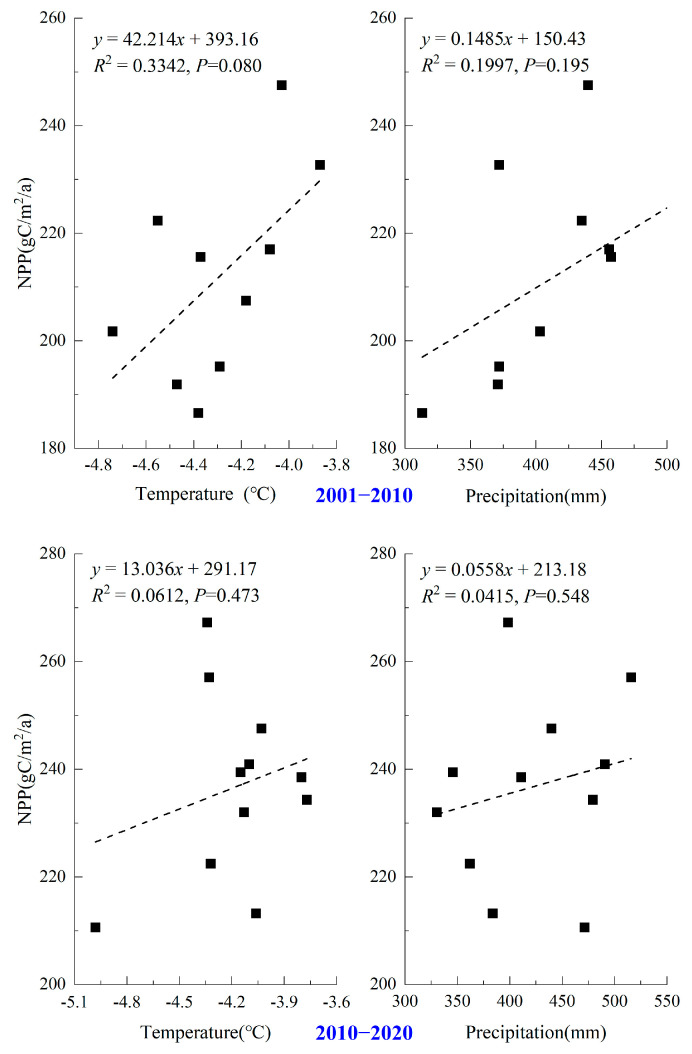
Scatter plot of net primary productivity (NPP), temperature, and precipitation in the Qinghai Lake Basin from 2001 to 2020.

**Figure 7 ijerph-20-02179-f007:**
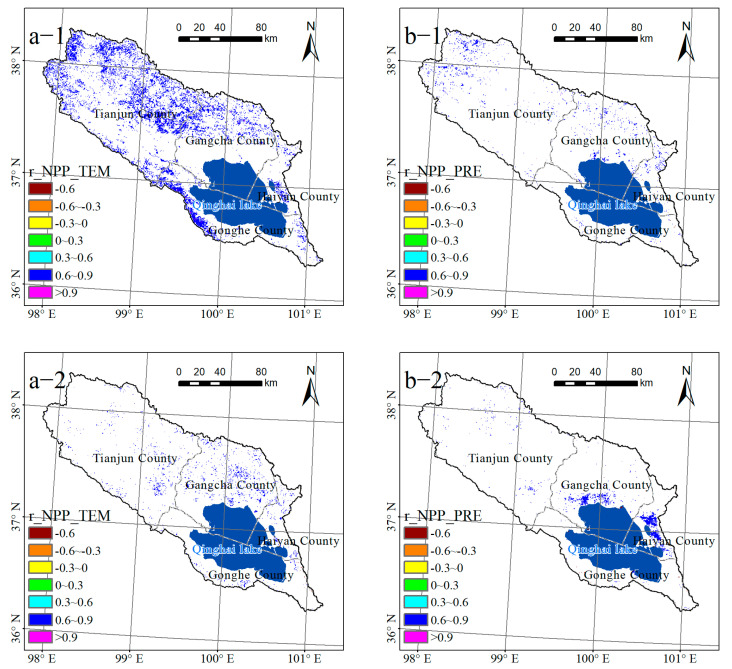
Partial correlation coefficients between net primary productivity (NPP) and temperature (**a-1**,**a-2**) and precipitation (**b-1**,**b-2**) in the Qinghai Lake Basin. (1 and 2, respectively, represent 2001–2010 and 2010–2020).

**Figure 8 ijerph-20-02179-f008:**
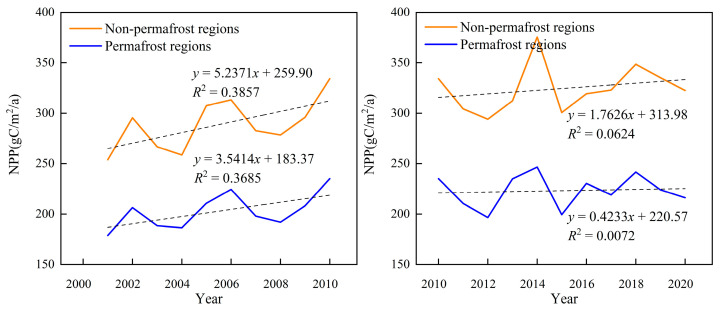
Trend of Net Primary Productivity (NPP) in permafrost and seasonal permafrost areas from 2001 to 2020.

**Table 1 ijerph-20-02179-t001:** Area and dynamic changes of land use/land cover (LULC) types in Qinghai Lake Basin from 2000 to 2020.

Type	Area/km^2^			LUCC Dynamics/%	
	2000	2010	2020	2000–2010	2010–2020
Cultivated land	451.41	448.48	447.11	–0.06	–0.03
Woodland	471.38	471.91	489.23	0.01	0.37
Grassland	18,599.93	18,561.83	18,514.35	–0.02	–0.03
Water bodies	5384.73	5339.16	5455.68	–0.08	0.22
Construction land	21.39	111.89	132.15	42.32	1.81
Unused land	4741.93	4737.77	4632.84	–0.01	–0.22
Comprehensive LUCC dynamics/%	\	\	\	7.78	3.04

**Table 2 ijerph-20-02179-t002:** Annual average net primary productivity (NPP) (gC·m^–2^·a^–1^) by land use/land cover (LULC) types at different time periods.

Year	LULC type			
	Cultivated Land	Woodland	Grassland	Unused Land
2001	277.83	291.56	192.24	158.04
2010	350.71	339.30	258.64	196.91
2020	363.54	319.56	244.00	174.88
Mean	330.69	316.81	231.63	176.61

**Table 3 ijerph-20-02179-t003:** Influence and contribution of different land use/land cover (LULC) types to the total NPP from 2001 to 2020.

LULC Type	Factors	2001–2010		2010–2020	
		Influence Quantity/Tg C	Contribution Rate/%	Influence Quantity/Tg C	Contribution Rate/%
Cultivated land	LUCC	−0.36	2.38	−0.21	8.69
	Climate change	14.46	96.99	2.19	91.03
Woodland	LUCC	0.07	0.64	2.77	55.97
	Climate change	11.02	99.24	2.10	42.47
Grassland	LUCC	−135.88	0.59	−227.89	4.21
	Climate change	22,940.71	99.21	−5165.52	95.54
Unused land	LUCC	−2.99	0.35	−94.17	15.15
	Climate change	851.48	99.56	−516.04	83.01

## Data Availability

The data presented in this study are available on request from the corresponding author.

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
