# Peer review of "Impacts of Climate Change and Land Use/Cover Change on the Net Primary Productivity of Vegetation in the Qinghai Lake Basin"

_ijerph, 2023, doi:10.3390/ijerph20032179_

Round 1

Reviewer 1 Report

This paper analyzes the effects of climate change and land use/cover change on net primary productivity of vegetation in Qinghai Lake basin. The idea is interesting and the result is basically satisfactory. I recommended publishing after minor revise. However, some other problems in the manuscript are still concerned in the following:

(1) The abstract is too descriptive. Revise it again.

(2) In Figure 1, the left figure is added with basic information such as climate zones and key place names to enrich the expression information of the figure.

(3) Increase the number of Landsat TM images used in 2.2.2.

(4) 2.3 Method description is too tedious. Please describe the main method.

(5) Figure 5 and Figure 7: increase the distribution of Qinghai Lake.

(6) The discussion on the impact of other factors on Net Primary Productivity in 3.3 lacks the support of key literature.

(7) Conclusion: Repeat the previous content and revise it again.

Author Response

Dear Reviewer 1: Thank you very much for your time consuming on our manuscript. Your suggestions are vital to improve the quality of our manuscript. I made quite a lot of necessary modifications to it according to the suggestions from you and other reviewers, please see them point by point in the following.

Thank you very much.

All the best regards,

Sincerely yours,

Jinlong Zhang and Hongwei Wang

Reviewer 2 Report

Thanks for all your work, this research is interesting and comprehensive. It is important to study the net primary productivity of vegetation and its driving force for the ecological stability of the dynamic region. The authors used CASA model, dynamic degree, LUCC transfer matrix, Sen+Mann Kendall and partial correlation analysis to analyze the impact of net primary productivity on vegetation (NPP) in Qinghai Lake basin, based on land use/land cover (LULC) data and meteorological data. I can see that the authors gave a significant effort to make the paper well_written and compelling. I recommend publishing this research in ijerph. To make the paper complete, here are a few questions and friendly suggestions. The details are as follows.

1. The Chinese expression mode of the article structure is very obvious, much like the writing method of a Chinese article. This requires the author to seriously consider refactoring.

2. The abstract of the article is the eye of the article. This abstract contains too much content. The author must introduce key points and innovations, rather than repeat the analysis of results.

3. The content of the introduction is obviously insufficient, and the research content is separated. For example, lines 45-50 describe the importance of NPP, but there is no expression related to the research content of the article. I strongly recommend that the author cite the reference: https://doi.org/10.1016/j.uclim.2022.101347, making the content of the article relevant, not only describing NPP but also considering LULC.

4. The 51-76 in the introduction introduces the research background and important aspects of NPP and LULC, but lacks adequate introduction and induction, and there is no recent research, such as GEE. I strongly recommend that the author cite these references in the introduction: https://doi.org/10.3390/land11081303, https://doi.org/10.3390/land11122281. In addition, any research is carried out on the basis of predecessors, and the article neglects to use up-to-date references as research support..

5.Some sentences are so long that it is hard to understand the sentences, I recommend that authors check the full text and refine the sentences. 

6.In the section of discussion, the prospect of the article is not deep enough, please provide future research directions with the limitations of this paper.

I believe the article will be greatly improved according to these suggestions. Good luck!

Author Response

Dear Reviewer 2:

Thank you very much for your time consuming on our manuscript. Your suggestions are vital to improve the quality of our manuscript. I made quite a lot of necessary modifications to it according to the suggestions from you and other reviewers, please see them point by point in the following.

Thank you very much.

All the best regards,

Sincerely yours,

Jinlong Zhang and Hongwei Wang

Reviewer 3 Report

The authors of the paper submitted to me for review examined an important topic regarding the impact of climate change on changes in land cover and land use and primary production in the lake catchment. The researchers used a rich set of remote sensing data, applied the methodology correctly, and the conclusion refers to the results obtained. I believe that the work deserves to be published in a journal with minor changes, namely:

Addition of country outlines in the location figure.

Specifying in the introduction what is a new achievement of this study in relation to previous works?

Development of the introduction.

Good luck!

Author Response

Dear Reviewer 3:

Thank you very much for your time consuming on our manuscript. Your suggestions are vital to improve the quality of our manuscript. I made quite a lot of necessary modifications to it according to the suggestions from you and other reviewers, please see them point by point in the following.

Thank you very much.

All the best regards,

Sincerely yours,

Jinlong Zhang and Hongwei Wang

Reviewer 4 Report

This manuscript assesses the impacts of climate and LULC on NPP in the Qinghai Lake Basin. Overall, the manuscript’s structure is good but its content needs much work. Please see below my comments and suggestions for the authors.

Line 22, spell out CASA like you did for other acronyms.

In the Introduction section, the literature review is rather thin. It should be further expanded to form a solid basis to justify your study, research question, research aims, and the main contributions of your study.

Line 67-68, you stated that “They focused on the effect of one or both of these factors on NPP and failed to consider the combined effect of multifactors.” It seems to me that you are implying your study would address the “combined effect”? If so, where and how it’s addressed?

Lines 72-76, first of all, this is a confusing statement. Is this the goal of your study? Or you are just expressing your view (if so, some references would be helpful).

Secondly, who’s influence are you referring to in your statement “…analyzing its influence on land use, climate change and other factors…”?

In Figure 1, add an inset map of China and highlight the location of your study area. This helps readers who are not familiar with China’s geography.

Lines 116 and 129, elaborate the reason why you decided to use the 500m resampling resolution.

Sections 3 and 4 require much attention. There is a disconnect between data, method, and result in sections 3 and 4. Please help your readers understand what data was used by what method, and what result was produced and how?

In the title and throughout the manuscript, the authors stated that climate change is a dominant factor that influences the changes in NPP. However, it’s unclear what climate change data was used, what analysis was performed, and how its impact on different LULC’s NPP was determined in Table 3?

If “There was essentially no link observed between the NPP and the annual mean temperature and precipitation from 2010 to 2020.” (lines 332-333), what is the basis for your statement “In conclusion, climate change was determined to be the dominant factor for the change in NPP in the Qinghai Lake basin.” (lines 309-310)?

Figure 7 a-1 vs. a2, what do you think of such a drop from 15.14% to 2.17% when it comes to temperature and NPP from 2001-2010 to 2010-2020? Why this might have happened?

Policy implications should be added to the Discussion section especially in the context, as you stated in lines 74-76 that “… to understand the driving mechanism of NPP evolution, ecological environment protection and sustainable development planning formulation have important theoretical and practical significance.”

Author Response

Dear Reviewer 4:

Thank you very much for your time consuming on our manuscript. Your suggestions are vital to improve the quality of our manuscript. I made quite a lot of necessary modifications to it according to the suggestions from you and other reviewers, please see them point by point in the following.

Thank you very much.

All the best regards,

Sincerely yours,

Jinlong Zhang and Hongwei Wang

Round 2

Reviewer 2 Report

The article has been improved, but it would be better if the introduction about LULC was further supplemented. Please refer to the following references. https://doi.org/10.3390/land11010014 . This supplement will further enrich the introduction of the article and enhance the interest of the topic.

In addition, the CO2 in line 37 should be CO2.

Author Response

(The authors gave the same response as above.)

Reviewer 4 Report

See my comments in the attached document.

Author Response

(The authors gave the same response as above.)
